# CPT Symmetry in Two-Fold de Sitter Universe

Davide Fiscaletti [1,*], Ignazio Licata [2,3] and Fabrizio Tamburini [4]

1 SpaceLife Institute, 61047 San Lorenzo in Campo, Italy
2 ISEM, Institute for Scientific Methodology, 90121 Palermo, Italy; Ignazio.licata3@gmail.com
3 School of Advanced International Studies on Applied Theoretical and Non-Linear Methodologies in Physics, 70121 Bari, Italy
4 ZKM—Zentrum für Kunst und Medientechnologie, Lorenzstraße 19, 76135 Karlsruhe, Germany; fabrizio.tamburini@gmail.com
* Correspondence: spacelife.institute@gmail.com

**Abstract:** The problem of baryon asymmetry unifies cosmology and particle physics at the hearth of theoretical physics. In this work, we consider the point of view of archaic cosmology based on the de Sitter hypersphere as topology of quantum vacuum. We show CPT symmetry derives from the nucleation of particles that divides the hypersphere in two mirror universes and defines big bang as a bifurcation point, as the creation of a de Sitter universe or a pair of entangled universes from "nothing". Then, we direct our attention to the behavior of neutrinos in a CPT universe and discuss the differences between Majorana and Dirac neutrinos in the observational imprints of the entangled universes.

**Keywords:** CPT symmetry; de Sitter Vacuum; entangled universes; Dirac and Majorana neutrinos



## 1. Introduction

The problem of baryonic asymmetry that we observe in the universe is one of the most formidable problems of theoretical physics and cosmology. A fundamental watershed was the pioneering work of A. Sacharov (1967) [1], which establishes three requirements that a theory of fundamental interactions must satisfy in order to explain bariogenesis: (a) violation of the baryonic number; (b) violation of C and CP or different decay rates between particle and antiparticle; and (c) processes out of thermal equilibrium. The implementation of these conditions in the Standard Model is a fascinating and complex path that introduces non-perturbing ingredients that constrain the Higgs potential [2]. The Standard Model potentially has all the characteristics to explain baryon asymmetry as a dynamic result starting from a symmetric solution, but from a quantitative point of view the results do not seem compatible with the experimental data. Recently, it has been shown that, in the class of theories with nonminimal curvature–matter couplings up to first order in the curvature, all the conditions of Sacharov without CPT violation can be naturally fulfilled [3].

An original and radical approach is proposed in a recent paper [4] to explain the fact that immediately after the big bang the universe can be associated to a spatially flat, radiation dominated FRWL metric. The idea is to extend CPT symmetry to the global structure of the universe. In this way, the universe after the big bang is the mirror image of the pre-big bang universe which implies a universe–anti-universe pair that emerges from the void and flows into a spatially-flat hot radiation-dominated era with FRWL metric. An interesting development is due to Volovik [5], who analyzed how the big bang emerges as a bifurcation point of a second order quantum transition from the Euclidean signature to the Minkowskian one, in which the symmetry between the two universes is spontaneously broken and any quantum superposition is eliminated. The results of the generalized CPT hypothesis are of great interest and far-reaching for the issue of dark matter and inflation.

Following Volovik's suggestion, it is legitimate to think that a generalized CPT can help clarify the age-old question of the transition between Euclidean signature with imaginary

time to the real-time one with FRLW metric, the conceptual core of the no boundary condition of Hartle-Hawking [6,7]. In this paper, we suggest a structural connection between CPT symmetry and de Sitter's archaic vacuum topology and the projective holography that extends the Hartle-Hawking Euclidean "semi-cone" with a hypersphere assumed as archaic vacuum geometry [8–10]. The work is structured as follows. In Section 2, we introduce briefly the model of archaic cosmology of de Sitter. In Section 3, we analyze CPT condition as a consequence of the topologic structure of the vacuum. The behavior of neutrinos is then considered in this scenario, in particular the Majorana neutrino which should be "indifferent" to the two mirror universes. In the conclusions, we suggest some working hypotheses based on these cosmological topologies.

## 2. The Archaic Universe as a Non-local Hypersphere

The model of the Archaic Universe is an approach to quantum cosmology developed on de Sitter-Arcidiacono projective relativity [11,12], which invokes an "archaic pre-space" as a pre-spatial and a-temporal substrate of the usual spacetime metric containing in nuce all the evolutionary possibilities that the General Projective Relativity equations indicate. In this model, after eliminating any geometrical singularity with Euclidean substrate, the description of the evolution of the universe can be seen as an extended nucleation from a coherent timeless state (de Sitter isotropic singularity) with very high non-local information to an observable mix of local matter-energy. For recent developments of this approach see, for example, the works in [8–10,13,14].

One of the traditional problems in working with de Sitter's universe is the ambiguous use of two representations, the hyperspherical one with imaginary and curved time and the other in the form of hyperbolic hyperboloid in real time, similar to a Sicilian "cannolo" (the most traditional of all Sicilian confectionery products). In both representations, the existence of a minimum radius avoids the singularity and sets a cosmological constant. It is interesting to note that the existence of a cyclic and imaginary time violates one of the Hawking-Penrose conditions of the singularity theorem [15].

Toresolve the question of the transition from imaginary to real time, in [8,9,14], L. Chiatti and one of the authors (IL) adopted the hypersphere S4 as the topology of the quantum vacuum. The model of the Archaic Universe postulates that S4 is an ancestor highly non-local phase with respect to the "big bang", in which the geometry of the universe is that of the four-dimensional surface of a hypersphere in the five-dimensional Euclidean space. The vacuum is therefore a Universal Action Reservoir of virtual processes [16] located on the four-dimensional surface of a five-dimensional Euclidean hypersphere.

To fix the ideas, the equation of the hypersphere S4 is the following:

$$(x_0)^2 + (x_1)^2 + (x_2)^2 + (x_3)^2 + (x_5)^2 = r^2 \tag{1}$$

The key link between geometry and physics is given by the axis $x_0$ of the hyper-sphere representing temperatures. This axis can be considered as an "archaic precursor" of time. To find the traditional big bang, let us assume that along the axis $x_0$ there is a critical value beyond which virtual processes are actualized in real matter. A reasonable value for bariogenesis is:

$$T_c = \hbar/k\theta_0 \tag{2}$$

where $\theta_0 \sim 10^{-23}$ s plays the role of the fundamental time interval (chronon), so $T_c$ is equal to approximately $10^{13}$ °K.

Considering the usual relationships between the probability of microstates and macrostates in statistical mechanics $P = \exp(-F/kT)$ and taking account of (2), we obtain:

$$P = exp(-Fx_0/\hbar c) = exp(-p_0 x_0/\hbar) = exp(-\Sigma/\hbar) \tag{3}$$

where the quantity F is the energy that the system would liberate if all the particles and fields which it is made of become real, $p_0 = F/c$, and $\Sigma$ is the total action held by the

Universe "before" the big bang. The following relation exists between the action and the entropy of the pre-big bang Universe:

$$\Sigma/\hbar = \text{-S/k} \tag{4}$$

In other words, $\Sigma$ is a negative entropy or, one might say, a sort of information whose bit is given by the quantity $\hbar\ln(2)$. From Equation (4), one has $-\Sigma = \hbar\ln(P)$ and, thus, for $P = \frac{1}{2}$ (binary choice), $\Sigma = \hbar\ln(2)$. In general, a dimensionless amount of information $I = \Sigma/[\hbar\ln(2)]$ can be introduced.

From the relation $\underline{x}_0 \leq c\theta_0$, which is valid in the "pre-big bang" era, if one puts $c\theta_0 = 2\pi\underline{R}$ one has $p^0\underline{x}_0 \leq 2\pi p^0\underline{R}$, i.e., $\Sigma \leq 2\pi F\underline{R}/c$. Thus, one obtains:

$$I \leq 2\pi F\underline{R}/[\hbar c\ln(2)] \tag{5}$$

and this is a form of the *Bekenstein relation*, which is valid for the "pre-big bang" phase [17].

An arc of a maximum circle perpendicular to the equator $x_0 = 0$ on the surface of the hypersphere is a seat of virtual processes which are interpreted as virtual fluctuations of "duration" $x_0/c$ (and, therefore, according to the uncertainty principle, of energy $\hbar c/x_0$). The hypersphere is therefore the timeless seat of the pre-vacuum. The formation of our universe constituted by ordinary matter can be described as a set of "nucleation" processes starting from a "universal reservoir" consisting of the virtual processes "written" on the four-dimensional surface, which is the timeless seat of the pre-vacuum. The crucial starting-point of our approach lies in Equation (2), which implies that the physical constraint for the bariogenesis is represented by the existence of a time interval $\theta_0$ such that for $\underline{x}_0/c > \theta_0$ the free energy of the pre-vacuum can be converted into real interactions between real elementary particles. The interval $\theta_0$ can be identified with the time required for light to travel the classic radius of the electron ($\approx 10^{-23}$ s); it identifies the particle scale, that is, the scale on which matter appears granular because it is made up of micro-events of interaction between "elementary particles".

In summary, in this approach, one can thus define two kinds of horizons of events, namely a cosmological horizon characterized by a fundamental constant of nature, $t_0$, which satisfies $ct_0 \approx 10^{28}$cm and is invariant in the cosmic time, and a microphysics horizon, where a localization of a particle or a reduction-state (R) process (following the Penrose terminology) occurs, which is defined by a radius $c\theta_0 \approx 10^{-13}$cm. The R processes characterized by the microphysics horizon $10^{-13}$cm are constituted of interaction vertices in which real elementary particles are created or destroyed and which are connected by the transformation of the same aspatial and atemporal pre-vacuum. According to this approach, therefore, the history of the Universe, considered at the fundamental level, can be seen as a complete network of past, present and future R processes deriving from the same invariant timeless pre-vacuum. It looks as if every quantum process and time itself emerge from this invariant timeless substratum and re-absorbed within it. The process of nucleation of real matter that "empties" the pre-vacuum, thus converting it into matter, is the big bang from the point of view of the space-time domain.

## 3. Hemispheres in the Pre-Big Bang Epoch

In [4], Boyle, Finn and Turok suggested that the state of the universe does not spontaneously violate CPT, that the universe after the big bang is the CPT image of the universe before it and that our invariant CPT-universe can be seen as a universe–anti-universe pair emerging from nothing. They found that, if one considers an FRW background equipped with an isometry under time reversal $\tau \rightarrow -\tau$, then there is a preferred vacuum that respects the full isometry group (including CPT); in other words, by imposing a CPT universe, one binds the vacuum. In this regard, these three authors considered a Weyl-invariant spinor field expressed by

$$\psi = a^{3/2}\Psi \tag{6}$$

where $\boldsymbol{\Psi}$ is the spinor with mass $>0$ on a flat FRW background and $a$ is the background scale factor. The field (6), which satisfies a Dirac equation of motion, can be expanded as

$$\psi = \sum_h \int \frac{d^3k}{(2\pi)^{3/2}} \left[ a_0(k,h)\psi(k,h,x) + b_0^+(k,h)\psi^c(k,h,x) \right] \tag{7}$$

where the operators $a_0$ and $b_0^+$ are particle and antiparticle operators defined as

$$\begin{bmatrix} a_0(k,h) \\ b_0^+(-k,h) \end{bmatrix} = \begin{bmatrix} cos\frac{\lambda(k)}{2} & \mp isin\frac{\lambda(k)}{2} \\ \mp isin\frac{\lambda(k)}{2} & cos\frac{\lambda(k)}{2} \end{bmatrix} \begin{bmatrix} a_\pm(k,h) \\ b_\pm^+(-k,h) \end{bmatrix} \tag{8}$$

in such a way that they transform as $[CPT]a_0(k,h)[CPT]^{-1} = -b_0(k,-h)$ and $[CPT]b_0(k,h)[CPT]^{-1} = -a_0(k,-h)$. This implies that the corresponding vacuum defined by $a_0|0_0\rangle = b_0|0_0\rangle$ is CPT invariant: $CPT|0_0\rangle = |0_0\rangle$. In the Boyle, Finn and Turok model, among the continuous family of invariant vacua defined by a real SO(2) rotation of $\left( a_\eta(k,h), b_\eta^+(-k,h) \right)$ through an angle $\eta$ satisfying $\eta(k) = -\eta(-k)$, the vacuum $|0_0\rangle$, which minimizes the Hamiltonian in the asymptotic $+/-$ regions regarding the solutions in the far future and in the far past respectively, is preferred.

In the approach developed in this paper of an archaic vacuum where particles emerge from nucleation processes, the CPT symmetry can receive a new explanation. In fact, while in the Boyle, Finn and Turok model the vacuum is bound by imposing a CPT universe thanks to the ad hoc assumptions (8) regarding the operators $a_0$ and $b_0^+$ which appear in the expression of the Weyl-invariant spinor field.

$$\phi(x,T) = \phi_0 exp\left[ \pm j\frac{2mE}{\hbar}x - j\frac{Et}{k} \right] \tag{9}$$

(where $j = \sqrt{-1}$), instead in our approach ad hoc assumptions are not necessary in the sense CPT symmetry emerges naturally in virtue of the features and physical dynamics regarding the Arcidiacono 5-sphere and its corresponding projective holography. In our approach, one can obtain a symmetry which is lacking in more conventional cosmology. In a pre-big bang phase, there is a fundamental symmetry between the two hemispheres into which the infinite-temperature equator divides the Arcidiacono hypersphere while after the big bang our Universe develops indeed from one of these two hemispheres into which the infinite-temperature equator divides the Arcidiacono hypersphere. This hemisphere is converted into the chronotope by means of a Wick rotation and the application of a scale reduction.

Instead, the other hemisphere does not play any role and can be considered as a simple mathematical artifact (it corresponds to the choice of a negative sign instead of a positive sign in the definition of the projective coefficient of the metrics [9,11,12,18]). This second hemisphere can be regarded as a second archaic universe that is mirror with our own and is characterized by archaic fluctuations exiting from the equator which correspond only to antimatter particles (while the vacuum fluctuations taking place towards our hemisphere correspond only to particles of ordinary matter). Thus, in the second hemisphere, a Universe would be developed in which antimatter would be dominant. This would take place in such a way that the CPT-symmetry of the archaic vacuum is maintained. These two "mutually mirror Universes" would be separated by the equator, and would therefore be causally unconnected, albeit contiguous. Their common origin would just be the archaic vacuum.

In this approach of two mutually mirror universes associated to each of the two hemispheres in the pre-big bang epoch, one can assume that within the antimatter hemisphere equation

$$t \to \frac{j\hbar}{kT} \tag{10}$$

holds with a sign minus at left hand (where $j = \sqrt{-1}$, $k$ is Boltzmann's constant, $\hbar$ is reduced Planck's constant and $T$ is real and positive or null). As a consequence, Equation (9) is now converted in the corresponding advanced wavefunction. In other terms, the fluctuation of energy $E > 0$, which before was converted in the retarded wavefunction

$$\boldsymbol{\phi}(\boldsymbol{x}, \boldsymbol{T}) = \boldsymbol{\phi}_0 exp\left[\pm j\frac{2mE}{\hbar}\boldsymbol{x} - j\frac{Et}{k}\right] \tag{11}$$

of energy $E > 0$, is now instead converted in the advanced wavefunction represented by the complex conjugate of (11), having an energy $-E < 0$. In this way, a negative energy can be associated with quantum fluctuations on the antimatter hemisphere. Due to the symmetry of the initial state at infinite temperature, this negative energy exactly compensates the positive energy of fluctuations on the matter hemisphere, so enabling a creatio ex nihilo. However, for an inertial observer exiting from the anti-big bang in the anti-Universe, which measures a growing cosmic time ($t \rightarrow -t$), the energy released at big bang in the form of antimatter is positive.

Although the entire proposal suggested here does not appear open to verification by observation, it nevertheless allows explaining the dominance of ordinary matter without introducing special initial conditions, at the same time preserving the greatest symmetry of the initial state in the sense that the CPT symmetry must not be postulated ad hoc. In other words, our approach allows us to explain the matter–antimatter discrepancy by obtaining CPT symmetry as a direct consequence of the features of the two hemispheres of the Arcidiacono hypersphere, the action of their advanced and retarded wavefunctions as well as their corresponding vacuum fluctuations.

Moreover, if the mechanism based on which ordinary matter and antimatter would seem to separate along the equator remains enigmatic, it appears plausible to presume that the particles acquire their own charges only at the time of the big bang, when the particles become real and therefore capable of real interaction and space becomes open.

If one assumes that the sign of the charges is defined by the hemisphere in which the particle becomes real (i.e., by the fact that it appears at the time of the big bang in our Universe or, alternatively, at the time of the anti-big bang in the anti-Universe) one does indeed obtain the required separation ab initio of matter and antimatter. The sign of the charge would in other words be defined by the direction of the timeline emerging from the equator along which the particle materialized, a result which could in some way be connected with the CPT theorem.

On the other hand, the fact—invoked by Volovik [5]—that the Big Bang emerges as the bifurcation point of the second-order quantum transition from the Euclidean to Minkowski signature, at which the symmetry between the spacetime and anti-spacetime is spontaneously broken, can also receive here a natural explanation. In fact, the change of sign of the scale factor around the big bang corresponds here to the behavior of the sign of the charges defined by the hemisphere in which the particle becomes real: the required separation of matter and antimatter is associated with the direction of the timeline emerging from the equator along which the particle materialized and this explains in what sense one obtains, after the bifurcation point represented by the big bang, a quantum transition from the Euclidean to the Minkowski signature, namely a spontaneous symmetry breaking between spacetime and anti-spacetime. In other words, the feature of the big bang as bifurcation point, which generates a spontaneous symmetry breaking and separation between matter and antimatter and the corresponding transition from Euclidean to Minkowski signature, emerges here as a direct consequence of the features of the two hemispheres of the Arcidiacono hypersphere, namely of the action of their advanced and retarded wavefunctions as well as their corresponding vacuum fluctuations. If in the picture proposed by Volovik the initial stage of the evolution of our Universe after the big bang can be characterized by the negative temperature, this is due to the fact that, in the light of Equations (9)–(11), a negative energy can be associated with quantum fluctuations on the antimatter hemisphere and, in virtue of the symmetry of the initial state at infinite temperature, this negative

energy exactly compensates the positive energy of fluctuations on the matter hemisphere. It is just the features of the fluctuations of the vacuum and of the corresponding advanced and retarded wavefunctions characterizing the two hemispheres of the Arcidiacono hypersphere which determine the action of the big bang as a bifurcation point, at a negative temperature, generating a spontaneous symmetry breaking between matter and antimatter. Moreover, as shown in [19], this idea of the correspondence between the action of the big bang as a bifurcation point and the quantum creation of an entangled pair of universe and anti-universe can be implemented in a slightly different way in terms of instantons [19] and the emergence of the classical spacetime suggests that the time variables of the two universes should be reversely related in a picture where quantum entanglement can exist between two causally disconnected regions in de Sitter space [20–22].

The thermodynamics of the arrow of time suggests that both branches of the Hartle-Hawking quantum state with no boundary conditions, describe an expanding universe by the Einstein-Hilbert action of a de Sitter-like spacetime originated by a Euclidean de Sitter instanton. An observer inside one of the entangled universes would see a universe in a thermal state where its properties are expected to depend on the properties of the entanglement. Observable consequences are expected, together with distinguishable imprints in the properties of our universe that would reveal that the process of creation of universes in universe–anti-universe pairs occurred. In fact, the direct non-observability does not exclude the possibility of measuring observable effects derived from the existence of quantum correlations or entanglement between the state of some matter field in two distant places. Thus, quantum entanglement affects the shape of the spectrum on large scales comparable to or greater than the curvature radius [23]. Traces of the entanglement of an initial state between two causally disconnected de Sitter spaces may remain also on small scales [24].

From this, one can formulate two main scenarios recalling the CPT archaic universe: the two universes may have been nucleated as an entangled pair of universes coherently under certain causal mechanism, and then they may have been separated off by the exponential expansion of space between us. They can remain separated in space and time, our twin partner universe may exist beyond the Hubble horizon and the quantum fluctuations of our universe may be entangled with those of the unobservable universe that could be detected as traces in the CMB of our universe (e.g., super horizons [25]).

On the other hand, another line of reasoning concerns the holographic approach and requires a subtle distinction between the global characteristics of de Sitter's hypersphere (Archaic Universe or bulk) and its real-time representations (screen, which rapidly expands driven by a small positive cosmological constant). In this regard, one can demonstrate that the peculiar global properties of AdS symmetry imply an entanglement entropy between the two four-dimensional screen components [26–29].

Finally, it is interesting to compare the approach here developed with a model suggested by Li-Xin Li [30] of a spacetime constituted by two open universes connected by an evolving Lorentzian wormhole, which satisfies the weak energy condition and is characterized by a negative spatial curvature, thus implying a continuous expansion and the absence of horizons and singularities in the future and the perspective that an observer can travel from one side to the other if he travels towards the future. A parallelism can be made between Li's model of a spacetime constituted by two open universes connected by an evolving Lorentzian wormhole and our approach of the archaic vacuum where particles appear as a consequence of nucleation processes in a picture of a CPT symmetry and the big bang emerges as a bifurcation point between the matter Universe and the antimatter Universe. If in Li's approach, the existence of a time arrow implies that an observer traveling from one side to the other of the wormhole will not feel any difference in local spatial geometry from the place where he starts and will not feel the existence of the "throat", since the wormhole expands so rapidly that as the observer passes through the wormhole to the other side he always sees that a two-dimensional cross-section in the FRW metric gets larger and larger; in an analogous way, in our model, we deal with

two hemispheres of the Arcidiacono hypersphere characterized respectively by matter and antimatter particles, where in a pre-big bang phase there is a fundamental symmetry between the two hemispheres while after the big bang our matter Universe develops indeed from one of these two hemispheres into which the infinite-temperature equator divides the Arcidiacono hypersphere. The idea of big bang as a bifurcation point between the two mirror universes, the matter Universe and the antimatter Universe, can be someway associated with the evolving Lorentzian wormhole with a time arrow of Li's approach.

## 4. CPT Symmetry and Majorana Neutrinos

In the approach of the archaic vacuum where particles appear as a consequence of nucleation processes in a picture of a CPT symmetry, the perspective is opened that Majorana neutrinos are the only particles which are the same in the two mirror universes, namely the matter universe and the antimatter universe, associated to each of the two hemispheres of the Arcidiacono Hypersphere, with a deep connection between the two universes in the twin universe model too. The physical time variables of the two universes must be reversely related and both universes are expanding universes with the observer's universe initially filled of matter and the partner universe initially filled with antimatter. The contracting and expanding branches describing the evolution of the twin universes would result equivalent for Majorana particles, as they are mirror particles and the two universes remain entangled or better with undistinguishable dynamics/evolution through these neutrinos [31].

Consider in the Archaic Universe the Majorana field $\psi_m$ that has a plane-wave expansion of the form

$$\psi_m(x) = \int \frac{d^3\vec{p}}{(2\pi)^{3/2}\sqrt{2E}} \sum_s \left\{ f\left(\vec{p}, s\right) u\left(\vec{p}, s\right) e^{-ip\cdot x} + \lambda f^+\left(\vec{p}, s\right) v\left(\vec{p}, s\right) e^{ip\cdot x} \right\} \quad (12)$$

where $f$ and $f^+$ represent the creation and annihilation operators for the Majorana particles of interest and $\lambda$ is a unimodular creation phase factor. By applying Equation (8), which defines a CPT invariant vacuum, one obtains the following transformations regarding $f$ and $f^+$

$$\begin{bmatrix} f\left(\vec{p}, s\right) \\ f^+\left(\vec{p}, s\right) \end{bmatrix} = \begin{bmatrix} cos\frac{\lambda(k)}{2} & \mp isin\frac{\lambda(k)}{2} \\ \mp isin\frac{\lambda(k)}{2} & cos\frac{\lambda(k)}{2} \end{bmatrix} \begin{bmatrix} f_\pm\left(\vec{p}, s\right) \\ f_\pm^+\left(\vec{p}, s\right) \end{bmatrix} \quad (13)$$

$$\text{CPT}\psi_m(x)(\text{CPT})^{-1} = -\eta_C\eta_P\eta_T\gamma^5\psi_m^*(-x) \quad (14)$$

where $\gamma^5 \equiv i\gamma^0\gamma^1\gamma^2\gamma^3$ and thus

$$CPTf\left(\vec{p}, s\right)CPT^{-1} = s\lambda^*\eta_C\eta_P\eta_T f\left(\vec{p}, -s\right)$$
$$CPTf^+\left(\vec{p}, s\right)CPT^{-1} = -s\lambda\eta_C\eta_P\eta_T f^+\left(\vec{p}, -s\right) \quad (15)$$

Since *CPT* is an antiunitary operator, one can write $CPT = KU_{CPT}$ where $U_{CPT}$ denotes a unitary operator. As a consequence, taking the Hermitian conjugate of either relation implies that $\eta_C\eta_P\eta_T$ is pure imaginary, and, thus, taking account of the results obtained in [32], the quantity $\eta_C\eta_T$ must be real. In contrast, the combination $\eta_C\eta_P$ is unconstrained. In summary, here one finds that all the restrictions on the phases that appear in C, P, T and combinations thereof are a consequence of the CPT invariance of the archaic vacuum.

The CPT invariant vacuum now throws new light into the oscillations of neutrinos invoked in order to provide a correct understanding of the expanding universe, recomposing the apparent mismatch between the motional energy of the galaxies and their gravitational energy, the former being an order of magnitude larger than the latter.

We now have clear experimental evidence that neutrinos are massive particles and there is mixing in the lepton sector. The matrix which is introduced to describe the mixing

is known as the Pontecorvo-Maki-Nakagawa-Sakata (PMNS) matrix, and its elements are now being measured with increasing precision by accelerator- and reactor-based neutrino experiments. On the basis of the treatment made in [33], by combining the full set of data, the following preferred ranges for the oscillation parameters is obtained:

$$\Delta m_{21}^2 = \left( 7.59 \begin{array}{c} +0.20 \\ -0.18 \end{array} \right) \cdot 10^{-5} eV^2, \ \Delta m_{31}^2 = \begin{cases} (2.45 \pm 0.09) \cdot 10^{-3} eV^2 \\ -\left( 2.34 \begin{array}{c} +0.10 \\ -0.09 \end{array} \right) \cdot 10^{-3} eV^2 \end{cases} \tag{16}$$

$$\sin^2 \theta_{12} = 0.312 \begin{array}{c} +0.017 \\ -0.015 \end{array}, \ \sin^2 \theta_{23} = \begin{cases} 0.51 \pm 0.06 \\ 0.52 \pm 0.06 \end{cases}, \ \sin^2 \theta_{13} = \begin{cases} 0.013 \begin{array}{c} +0.007 \\ -0.005 \end{array} \\ 0.016 \begin{array}{c} +0.008 \\ -0.006 \end{array} \end{cases} \tag{17}$$

where $\Delta m_{ij}^2 = m_i^2 - m_j^2$ are the mass squared differences between the neutrino mass eigenstates $\nu_{i,j}$; $\theta_{ij}$ are the corresponding angles in the standard three-flavor parameterization of the neutrino mixing matrix $V_L$ [34]; and the upper (lower) rows correspond to normal (inverted) neutrino mass hierarchy.

Non-zero neutrino masses generated inside a CPT archaic vacuum introduce the suggestive perspective of a new physics beyond the Standard Model. Right-handed neutrinos are an obvious possibility to incorporate Dirac neutrino masses. However, the $\nu_{iR}$ fields would be $SU(3)_C \otimes SU(2)_L \otimes U(1)_Y$ singlets, without any Standard Model interaction. If such objects do exist, it would seem natural to expect that they are able to communicate with the rest of the world through some still unknown dynamics. Moreover, the Standard Model gauge symmetry would allow for a right-handed Majorana neutrino mass term,

$$L_M = -\frac{1}{2}\overline{\nu}_{iR}^c M_{ij}\nu_{iR} + h.c. \tag{18}$$

where $\nu_{iR}^c$ denotes the charge-conjugated field and $M_{ij}$ is the Majorana mass matrix which is not related to the ordinary Higgs mechanism. Since both fields $\nu_{iR}$ and $\overline{\nu}_{iR}^c$ absorb $\nu$ and create $\overline{\nu}$, the Majorana mass term mixes neutrinos and anti-neutrinos, violating the lepton number by two units, thus introducing the perspective of building a new physics in the context of the CPT archaic vacuum.

In fact, if one considers the CPT invariant archaic vacuum proposed in this paper, without any assumption about the existence of right-handed neutrinos or any other new particles, one obtains hints towards a new physics by considering the following general $SU(3)_C \otimes SU(2)_L \otimes U(1)_Y$ invariant Lagrangian

$$\Delta L = -\frac{c_{ij}}{\Lambda}\overline{L}_i s_R \widetilde{s}^t L_j^c + h.c. \tag{19}$$

where $\Lambda$ is the new-physics scale; $L_i$ denotes the i-flavored $SU(2)_L$ lepton doublet, $\widetilde{s}_R = i\sigma_2 s_R^*$, $s_R$ being the singlet field describing the occurrence of creation/annihilation of the sub-particles of the archaic vacuum; and $L_i^c$ is the charge-conjugated i-flavored $SU(2)_L$ lepton doublet. After spontaneous symmetry breaking, one obtains $\left\langle s_R^{(0)} \right\rangle = \lambda_R/\sqrt{2}$, where $\lambda_R$ is the coupling associated with the real part of the singlet field $s_R$ (which, in the spontaneous symmetry breaking regime, has the expression $\lambda_R = \frac{75}{g\sin\theta_W}$ where $g$ is the coupling of the isospin current $\overrightarrow{J}_\mu$ of the fermions to the massless isovector triplet $\overrightarrow{W}_\mu$—for $SU(2)_L$—which appears in the original Weinberg–Glashow–Salam model) and $\Delta L$ generates a Majorana mass term for the left-handed neutrinos, with $M_{ij} = c_{ij}\lambda_R^2/\Lambda$. Therefore, in the light of Equation (19), one can say that the Majorana mass term for the left-handed neutrinos are ultimately connected by the coupling of the real part of the singlet field $s_R$. Taking $m_\nu \geq 0.05 eV$, as suggested by atmospheric neutrino data, here one gets $\Lambda/c_{ij} \leq 10^{15} GeV$, close to the expected scale of Grand Unification.

As a consequence of non-zero neutrino masses, the leptonic charged-current interactions contain a flavor mixing matrix $V_L$, which thus may be considered as a consequence of the elementary creation/annihilation events in the CPT archaic vacuum, and they in turn lead to the appearance of the "bare" mass of the particles and the functions describing them. As is known, the data on neutrino oscillations imply that all elements of $V_L$ are large, and this means that the mixing among leptons appears to be very different from the ones in the quark sector. We can also suggest that these features of the elements of the mixing matrix $V_L$ and of the consequent mixing among leptons derive from the functions describing the elementary creation/annihilation events in the CPT archaic vacuum. The number of relevant phases characterizing the matrix $V_L$ depends on the Dirac or Majorana nature of neutrinos, and, with only three Majorana (Dirac) neutrinos, the $3 \times 3$ matrix $V_L$ turns out to involve six (four) independent parameters: three mixing angles and three (one) phases, and in our model all these are connected with the elementary processes of creation/annihilation, R processes of the CPT archaic vacuum.

Although recent experiments are exploring the sensitivity to new physics scales (such as the MEG experiment, which is searching for $\mu^+ \to e^+ \gamma$ events with a sensitivity of $10^{-13}$ [35]; ongoing projects at Fermilab [36,37] and J-PARC [38] aiming to study $\mu \to e$ conversions in muonic atoms at the $10^{-16}$ level; and proposals to reach sensitivities around $10^{-18}$ [39]) and although we have various data regarding the violation to CP invariance and the Majorana nature of the neutrino [40,41], which seem to indicate that the allowed values for the Majorana effective mass turn out to be < 75 meV at 3σ C.L. and lower down to less than 20 meV at 1σ C.L, we can conclude that, at present, we still ignore whether neutrinos are Dirac or Majorana fermions. Another important question to be addressed in the future concerns the possibility of leptonic CP violation and its relevance for explaining the baryon asymmetry of our matter Universe through leptogenesis. In fact, it is not yet known whether CP violation occurs in the neutrino sector. The low level of CP violation possible from Standard Model mechanisms has not been reconciled with the gross violation inherent in the fact that the observable universe seems to consist overpoweringly of matter rather than antimatter. Further research will provide more information about these topics. Here, it is interesting to mention that hints of new physics beyond the Standard Model towards a CPT universe emerging from an archaic vacuum have appeared recently also about neutrinos, with convincing evidence of neutrino oscillations showing that $\nu_i \to \nu_j$ (with $i \neq j$) transitions take place.

The existence of lepton flavor violation opens a very interesting window to unknown phenomena. In summary, the smallness of neutrino masses suggests new physics at very high energies, close to the expected scale of Grand Unification, in the context of a CPT archaic vacuum.

## 5. Perspectives: Micro and Macro CPT

The CPT problem has developed by increasingly intertwining particle physics and cosmology. Any attempt to find a solution must therefore confront its counterpart. Within the cosmological theory outlined here, we trace a possible connection with the physics of neutrinos through a model of decoherence and dissipation. In this regard, if several authors made many efforts to study dissipation and its origin in neutrino oscillations (see, e.g., [42–55]), in the recent papers [56,57] on the other hand, the time evolution of the density matrix for neutrinos is analyzed, finding that the presence of Dirac and Majorana phases in the mixing matrix generate, for an off-diagonal dissipator, in the three-flavor mixing case, a breaking of the CPT symmetry. The authors have found that, for a simple off-diagonal dissipator, Majorana neutrinos can violate CP symmetry in all the flavor preserving neutrino transitions because of the presence of three phases (the Dirac phase and the two Majorana phases) in the mixing matrix, while Dirac neutrinos can break CP symmetry only in two of the three flavor preserving transitions. Moreover, the authors have evidenced the possibility for Dirac neutrinos of the existence of a CPT violation due

to the Dirac phase $\delta$, while for Majorana neutrinos of the dependence of the oscillation formulae on the Majorana phases $\phi_i$, which generate themselves a CPT violation term.

Now, in the model of archaic vacuum, the decoherence and dissipation effects into the oscillations of neutrinos with their consequent violation of CP and CPT, may be seen as a final result of the separation of matter and antimatter which takes place with the bifurcation point represented by the big bang. According to our point of view, the key of explanation lies just in the spontaneous symmetry breaking between the spacetime and anti-spacetime and the corresponding emergence of the direction of the timeline from the equator along which the particles of our matter universe materialized, which are a direct consequence of the features of the two hemispheres of the Arcidiacono hypersphere, namely of the action of their advanced and retarded wavefunctions as well as their corresponding vacuum fluctuations. In other words, we suggest that the spontaneous symmetry breaking between the spacetime and anti-spacetime and the corresponding emergence of the direction of the timeline appear in our matter universe in the form of decoherence and dissipation effects, which lead to processes of violation of CP and CPT regarding the oscillations of Dirac and Majorana neutrinos at our upper level of physical reality.

For Dirac neutrinos, anyway, one then has clear observational imprints present in the spectrum of fluctuations of the matter field for the horizon modes. The large modes are in the vacuum state and then do not feel the inter-universal entanglement [31].

The CPT symmetry breaking in Dirac and Majorana neutrinos caused by decoherence and dissipation effects and the link with the recent results obtained in [56] suggest interesting developments.

Another proposal comes from axion physics where the cosmological excess of baryons over antibaryons is generated from the rotation of the QCD axion [58]. The complex relationships between elementary particles, which have played a key role at the origin of the universe [59–75] and the relative experimental difficulties, make a clear picture of the fine-tuned primordial mechanisms far away. However, we can say that a unified framework of the micro CPT issue (decoherence, dissipation and axion rotation) must be revealed at a cosmological level in a de Sitter model where the big bang turns out to be geometrically a bifurcation point.

**Author Contributions:** Data curation, D.F., I.L. and F.T. writing—original draft preparation, D.F., I.L. and F.T. writing—review and editing, D.F., I.L. and F.T. All authors have equally contributed to this work. All authors have read and agreed to the published version of the manuscript.

**Funding:** This research received no external funding.

**Institutional Review Board Statement:** Not applicable.

**Informed Consent Statement:** Not applicable.

**Data Availability Statement:** Not applicable.

**Acknowledgments:** The authors thank Leonardo Chiatti (ASL VT Medical Physics Laboratory, Via Enrico Fermi 15, 01100 Viterbo, Italy) for the valuable suggestions on the work and the long and friendly collaboration cruising around at cosmological frontiers. FT thanks Peter Weibel and ZKM for the financial support.

**Conflicts of Interest:** The authors declare no conflict of interest.

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
