# Peer review of "CPT Symmetry in Two-Fold de Sitter Universe"

_symmetry, doi:10.3390/sym13030375_

Round 1

Reviewer 1 Report

The paper discusses the important and interesting points concerning the origin of Big Bang and the asymmetry between matter and antimatter. The authors propose a specific model which is based on some archaic models such as the DeSitter universe and the Arcidiacono two hyperspherical model composed of two hemispheres. The role of the CPT symmetry is analyzed and the connection between the cosmological process like Big Bang and pair creation in the mycroworld is pointed out. The paper is written very clearly and presents important and timely subject, therefore I recommend its publication

Author Response

Dear referee,

we have prepared a revised version of the paper according to your comments.

Regards,

Davide Fiscaletti

Reviewer 2 Report

Report on manuscript symmetry-1094721 Fiscaletti:

Fiscaletti and his coworkers attempted to address the question of the Euclidean signature involved with imaginary time or real time by generalizing CPT symmetry. Their resolution to this question is to introduce a structural connection between the CPT symmetry and the de Sitter's archaic vacuum topology by extending the Hartle-Hawking's Euclidean semicone with a hypersphere. To do this, the authors started with a quantum cosmology model, the Archaic universe developed upon the de Sitter-Arcidiacono's projective relativity and treated as a nonlocal hypersphere. Then they considered the CPT as the nucleation of particles by separating the hypersphere into two mirror universes and treated the big bang as a bifurcation point (or a pair of entangled universes). After these, they then looked at the neutrinos, and pointed out the Majorana neutrinos would behave differently in the two mirror universes.

After going through the work, I found the manuscript was not well written or organized. The presentation is hard to be understood and followed, especially due to the language and grammar issues. I also found the manuscript is misleading and easily gets people lost.

Here are some critical issues appearing in the content:

1.  Superficial discussions. The discussions are shallow and not rigorous. The authors seemed to grab different pieces of knowledge and put them into a box. Many key concepts involved are lack of rigorous definitions. It seemed that the different topics were only loosely linked together. Solid and concrete connections should be further developed and stated in a more precise way.

2.  Gaps. In the introduction of their structural model, the authors developed their idea by resorting to the verbal descriptions instead of mathematical proof, which would make people lost. 

3.  What does "sen" represent in Equations (8) and (13)?

4. Many symbols were lack of explanations or not defined. 

5.  Why should the quantities be these in Equation (17)? Any justification to the current work?

In short, I found I got lost in reading this manuscript. I strongly suggest the authors to make an extensive revision before a further decision could be reached.

Author Response

Dear referee,

We thank the referees for the careful analysis they wanted to dedicate to our work that helped us to improve it. The resubmitted version contains red highlights that show where we improve the text and answer the questions raised.

Please, find below our replies to reviewers, point by point:

1.  Superficial discussions. The discussions are shallow and not rigorous.
The authors seemed to grab different pieces of knowledge and put them into a box. Many key concepts involved are lack of rigorous definitions. It seemed that the different topics were only loosely linked together. Solid and concrete connections should be further developed and stated in a more precise way.

Reply : In our opinion, this is an impression of the referees that we respect, but it does not seem motivated to us, and this objection seems to us not a detailed criticism.
Anyway, we can guess from where this objection could come from. After our contribution published in Physica Scripta [CPT symmetry in cosmology and the Archaic Universe, May 2020, Physica Scripta 95(7); for a preprint version see: https://arxiv.org/abs/2002.07550], based on the concept of microevent of localization, our intention was to describe the same scenario from the point of view of the hyperspheric universe and its two "semicones", an issue on which confusion is often made, unfortunately, and leaving in the background of our discussion the Event Based Quantum Mechanics, published in Symmetry [https://www.mdpi.com/2073-8994/11/2/181 ].
We would like to point out that the Archaic Universe, despite its eccentric name, is a type of quantum cosmology developed from Hawking's ideas [see: F.Feleppa, I.Licatab, C.Corda: Hartle-Hawking boundary conditions as Nucleation by de Sitter Vacuum in Physics of the Dark Universe, Volume 26, December 2019, 100381: https://arxiv.org/abs/1909.07824 ]. We want to point out also that these are not heterogeneous ingredients : in De Sitter's Archaic Universe the cosmological wave function is the product of individual locations. From a practical point of view for a practical purpose, it is equivalent to QFT.

2.  Gaps. In the introduction of their structural model, the authors developed their idea by resorting to the verbal descriptions instead of mathematical proof, which would make people lost

Reply: please see the reply to pint 1. We wanted to focus mainly on the conceptual aspect, but in par. 2 there are all the essential aspects of the local/non-local relationship.

3.  What does "sen" represent in Equations (8) and (13)?
Reply: it was sin, now corrected.

4. Many symbols were lack of explanations or not defined.
Reply: The symbols have been explained in the text now.

5.  Why should the quantities be these in Equation (17)? Any justification
to the current work?
Reply: please, see ref. 33. see ref. 33. We looked for neutrino discriminating behaviors in the "twofold" universe, but it seems that we live in a "Majorana Universe". This is very interesting in our opinion.

Regards,

Davide Fiscaletti, Iganzio Licata, Fabrizio Tamburini

Reviewer 3 Report

Sections 2 and 3 should be improved - -recommendations are in the attached Report

With taking into account changes in Sections 2-3, the rest of the paper should be properly edited including language, style, and logic.

Author Response

(The authors gave the same response as above.)

Round 2

Reviewer 2 Report

In the revised manuscript, the authors clarified the confusions I had in the previous report. In particular, from the response I now understood the authors goals in this work. Although they aimed at presenting a  conceptual idea, it still would be helpful if the materials in the work could be connected smoothly. Nevertheless, I would support its publication in Symmetry with this revised version.